

# Potential of seasonal hydrological forecasting of monthly run-off volumes for the Rhone and Arve Rivers from April to July

Oriane Etter[1], Frédéric Jordan[2], Anton J. Schleiss[1]

[1]Laboratory of Hydraulic Constructions (LCH), EPFL, Lausanne, 1015, Switzerland
[2]Hydrique Engineers, Mont-sur-Lausanne, 1052, Switzerland

*Correspondence to*: Frédéric Jordan (fred.jordan@hydrique.ch)

**Abstract.** In a context where water management is becoming increasingly important, reliable seasonal forecasting of discharge in rivers is crucial for making decisions several months in advance. This paper explores the potential of seasonal forecasting of run-off volumes produced by ensemble streamflow forecasting using past climatology and comparing it to the
more commonly used average of past discharge measurements. The seasonal forecast was obtained for the Arve and Rhone rivers by simulation using the *Routing System* model for lead times of 30, 90 and 120 days. The initialization was performed on a validated simulation of 12 and 16 years for the Arve and Rhone rivers, respectively, obtained through long-term calibration. The performance was assessed by indicators called "accuracy" and "thinness". The normalized mean average error (NMAE) was used to compare the performance of the seasonal forecast with the average of the past measurements.
After a bias correction of the seasonal forecast of the Rhone River with the observed run-off volumes during the different lead times, the correlation of the median forecast with the measurements (accuracy) was larger than 0.55 for all lead times from April to July. The Arve River's accuracy was improved by disregarding the year 2007 member, leading to the floods of the 3rd and 9th of July, for lead times of 90 and 120 days. This resulting in the period of April to July having correlation accuracies higher than 0.5. For both rivers, the 80 % confidence interval of the seasonal forecast was relatively thin
compared to the measurements (thinness) for the months of April to July. The NMAE was used to validate the range of validity of the forecast. The correction of the forecast resulted in more months being favorable for seasonal forecasting for the Rhone River. The post-processing on the Arve River decreased the difference between the measurements and the forecast (NMAE). Further investigation should concentrate on dividing the meteorological datasets to produce a strong median forecast and confidence interval.

**Copyright statement**



# 1 Introduction

Optimizing water management will become increasingly important as alpine climates shift towards less precipitation falling as snow in the winter months, higher temperatures leading to more evapotranspiration, and more melt water in the spring, reducing the winter water storage in snowpack and glaciers (Beniston, 2012; Berghuijs et al., 2014; Middelkoop et al., 2001). This will impact all the actors relying on water, such as hydropower, water supply for agriculture and households, protection against floods and droughts, and artificial snow production in ski resorts (Céron et al., 2010; Schepen et al., 2016; Yuan et al., 2015b). Seasonal forecasting provides forecast lead times up to several months in advance, which will be of great advantage for the above-mentioned sectors. Nevertheless, seasonal forecasting has not significantly evolved in the past few years, and the challenge is still to have reliable forecasts with a lead times of more than one month (see Pagano et al. (2004), as quoted in Shi et al. (2008)).

Seasonal forecasting can be produced using two approaches, namely, the numerical or dynamic approach, using model simulations, and the empirical or statistical approach, using multiple regression, neural networks, a.s.o. (Murphy et al., 2001). Statistical methods rely on observed data, which allows one to identify trends or to build datasets for training algorithms. The use of statistical forecasting is widespread in the hydropower industry due to the amount of available data. Nevertheless, hydrological processes are often not well understood, and a seasonal forecast is simply an average of past observed discharges. Although simple, the purely statistical methods cannot account for initial snow pack or the relevance of the historical discharge compared to the period of the forecast.

The numerical approach is based on a good understanding of physical processes, which makes it very powerful. Ensemble forecasting is a numerical approach that allows the transition from deterministic to stochastic forecasting. It was first investigated for meteorological modeling. Ensemble forecasts use sets of initial conditions combined with sets of climatology to compute possible future scenarios over large lead times (Crochemore et al., 2017). The high nonlinearity of climate models makes seasonal meteorological forecasting challenging. The use of coupled atmosphere-ocean-land general circulation models helps improve seasonal forecasting through predictors that are relatively well understood, such as the North Atlantic Oscillation (Bierkens and van Beek, 2009) and the El Niño Southern Oscillation (Yuan et al., 2015a). The extension of forecast lead times is also of great interest in hydrological modeling. The seasonal predictability of hydrological models has been shown to be higher than that of climatic models (Céron et al., 2010), revealing their high potential for seasonal forecasting.

The use of ensemble streamflow prediction (ESP) was first described by Day (1985). The idea was to spin up a deterministic model with observed meteorological forcing up to the date of initialization of the forecast, then use a set of historical meteorology data as a forcing ensemble to produce different stream flow forecasts. Spinning up the calibrated model to the





initialization date should reproduce the hydrologic state by accounting for the soil moisture content and the glacial and snow storage (Day, 1985; Shi et al., 2008), providing realistic initial conditions for the seasonal forecast. The seasonal forecast is then run for the chosen lead times. The ensemble approach is capable of producing multiple forecasts from which quantiles can be extracted. The probabilistic information, which is associated with the quantiles, makes it more accurate than a single

forecast for decision making (Hagedorn et al., 2005). Additionally, while the statistical approach is widespread and reliable, the deterministic approach can account for changing trends, which has more potential for future forecasting (see Zwiers and Von Storch, 2004, as quoted in Shi et al., 2008; Yossef et al., 2017).

When using ESP, errors can originate from parameterization, initial conditions and meteorological forcing (Yossef et al.,

2017). In traditional ESP, errors due to forcing or initialization are considered negligible due to calibration (Mendoza et al., 2017). Schick et al. (2017) showed that for catchments with snow accumulation, the choice of initial conditions is crucial. To quantify the error due to the initial conditions, reverse ESP can be used. Reverse ESP consists of spinning up the model with randomly shuffled climatology to produce a set of initial conditions and forcing it with observed meteorology (Li et al., 2009). It is distinguished from traditional ESP, where the model is spun up to the date of initialization with observed

climatology. Comparing the influence of the forcing errors and the initial conditions errors, Li et al. (2009) showed that up to one month, the error in the initial conditions dominates, and for more than one month, the error in forcing will become dominant. Yossef et al. (2017) showed that the forecast from July to October in Western Europe is more sensitive to errors in forcing than in initial conditions.

Due to these uncertainties in initial conditions and model calibration, Shi et al. (2008) investigated the potential of post-processing the seasonal forecast to correct for the errors in initial conditions without the need for calibration. They found that the reduction in forecast error by post-processing of the forecast is nearly as good as the reduction obtained by model calibration. For ESP to be more reliable, either a bias correction of the forcing can be applied by pre-processing to reduce errors (Crochemore et al., 2016, 2017) or the forecast can be post-processed to reduce the errors due to forcing and

initialization. Their results showed that different methods of post-processing the forecast improve different performance indicators depending on the method chosen. In this study, the choice was to calibrate the model over a long period due to the presence of glaciers and to consider the initial conditions as perfect, leaving the errors to post-processing.

The traditional ESP methodology takes historical climate as the meteorological forcing for the deterministic models up to the

time of initialization, and it randomly samples the historical meteorology to forecast the future (Li et al., 2009). This method targets the forcing uncertainties. Some weather services provide seasonal meteorological forecasts, such as the Climate Forecast System version 2 (CFSv2) by NOAA's National Centers for Environmental Prediction (NCEP) or the ECMWF System 1−4, by the European Center for Medium-Range Weather Forecasts (ECMWF). The use of these meteorological ensembles in this contribution was disregarded based on the results of Crochemore et al. (2016), Lucatero et al. (2017b) and



Yuan et al. (2011). In all contributions, the improvement to the hydrological seasonal forecast through the use of meteorological seasonal forecasts depended on the location of the basin, the time of year and the lead time. Yuan et al. (2011) showed that CFSv2's performance was interesting for a one-month horizon, which is shorter than the lead time considered in this study. Lucatero et al. (2017b) tried using ECMWF System 4, but the accuracy of the ESP with observed forcing was higher, even with bias correction (post-processed) applied to the mean, minimum and maximum. Finally, MeteoSwiss (Swiss Federal Office of Meteorology and Climatology) provides a qualitative seasonal forecast, but it is difficult to use as an input or for performance comparison.

The main problem faced by traditional ESP is the assumption that the measured climatic data are representative of the climate during the forecasted period (Lucatero et al., 2017a, 2017b; Mendoza et al., 2017). Recently, ensembles of meteorological forcing have tended to be reduced to relevant years or are modified by weighing (Crochemore et al., 2016, 2017).

Seasonal forecasts can produce different types of results, such as varying discharge with time, volume for a period, soil moisture content, and average discharge over a period. For example, in drought management, the forecast is designed to produce a distribution of the discharge with time in order to know when to expect what amount of water (Schepen et al., 2016). This relatively high time resolution is not necessarily needed, especially for large reservoirs. In the case of large lakes, which are level regulated, the volume produced at a larger time resolution is sufficient. Changing the *predictand* (what is to be predicted) from a discharge distribution to volume simplifies the approach, because the climatology can be considered independent of the succession of daily events, depending solely on the initial conditions and the climatology. Thus, the reshuffling of past climatology becomes irrelevant.

This paper describes and discusses the use of ESP to forecast seasonal run-off volumes, without reshuffling the forcing, for two rivers whose management plays a major role in the management of Lake Geneva, not only through the optimization of electricity production but also by providing drinking water to the different cities around the lake and preventing floods.

The response time between rainfall and runoff is often a problem in seasonal forecasting for long lead times, due to the hydrological response of the basin being shorter; therefore, the presence of lakes can increase predictability (Sene et al., 2017). Evaluating which inputs to take into account over a lake can become significant depending on lake size. Sene et al. (2017) showed that for Lake Victoria and Lake Malawi, the water balance could be determined based solely on precipitation, and they showed that the skill in the level forecast was up to 3−6 months.



A seasonal skill is expected for catchments where the melt water dominates the discharge pattern of the spring and summer and where the soil moisture content guides the flow in autumn and winter. Pluvio-nival catchments should be good targets to find these skills.

The current paper aims to assess the skill of seasonal forecasted volumes produced by the *Routing System* (RS) model in comparison with the statistical predicted volumes (average of historical measurements) for the Arve and Rhone river basins.

## 2 Study domain

The two river basins studied are the Rhone River catchment in Switzerland upstream of Lake Geneva and the Arve River in the French Alps, a tributary of the Rhone River downstream of Lake Geneva . Their characteristics  can be found in Table 1
and their locations in Fig. 1. In addition to drinking water services, both catchments are involved in the management of Lake Geneva. Variation in the lake's level was the cause of several flooding events in the City of Geneva and other localities around the lake in the past. Although situated downstream of the lake, the Arve River is the main hydraulic limitation to the outflow of the lake (Grandjean, 1990). Since 1884, Lake Geneva's level has been regulated by a gated weir combined with a power plant (Grandjean, 1990; Lang et al. 1990). Since the 70s, the water management has been governed by the
optimization of the energy production as well as the guarantee of sufficient flow downstream to the different industries, such as the cooling systems for the nuclear power plants in France (Grandjean, 1990). The catchment of the Arve River contributes not only to the discharge of the Rhone River downstream of Lake Geneva but also upstream to the Rhone River entering Lake Geneva. Some water from the Chamonix glacier in France is diverted through tunnels towards the Emosson dam in Switzerland. Nevertheless, France has the right to use this water. Other rivers are tributaries of the lake but are not
considered in this study, since the Rhone River is responsible for three-fourths of the inflows into the lake, with only 8 % of the total inflow coming from precipitation on its surface (Grandjean, 1990). These rivers, the Veveyse, Venoge, Aubonne, Promenthouse, Dranse and Foron rivers, will be considered in the future operational implementation of seasonal forecasting for the inflows.

Compared with the Arve River, which only has three water infrastructures influencing its discharge, 46 large hydropower plants are located on the Rhone catchment (Etat du Valais, n.d.). In alpine rivers, the discharge, in principal, is dominated by the diurnal and seasonal cycles of snowmelt with the contribution of rainfall (Mutzner et al., 2015; Nolin et al., 2010; Schaefli et al., 2007). Nevertheless, with all the hydraulic infrastructures, such as hydropower plants and reservoirs, present in the catchment, the discharge undergoes a very different diurnal, weekly and seasonal pattern due to the energy production.
The daily discharge pattern downstream of storage power plants mostly depends on the prices of electricity and is difficult to model. Thus, the volume should be reproduced by the model on the monthly scale. The impact of the hydroelectricity is a shift from the natural seasonal pattern with smaller discharges in the summer (accumulation in reservoirs) and higher flow in



the winter (production from the reservoirs) (Grandjean, 1990). The measured and simulated discharge of the Rhone and Arve rivers can be seen in Fig. 2. The volume ratio of the calibration is 0.99 for the Rhone River and 1.04 for the Arve River.  The Rhone has a flow regime dominated by snow and glacial melt in the spring and summer with higher discharges. The summer discharge reaches three times that of the winter discharge for both catchments, but the Rhone River has much larger

discharges due to a much larger catchment area.

## 3 Methodology

The seasonal forecast is computed by the model *Routing System* (RS) with the module RS HYDROPOWER (Jordan, 2007; Schaefli et al., 2005) (Fig. 3). The models of the Rhone River and Arve River are already used as operational models for shorter lead time forecasts (Hydrique Engineers, Mont-sur-Lausanne, 1052, Switzerland). The model of the Rhone River was

described by Hernández et al. (2009).

RS is a deterministic rainfall-runoff model that was initially developed at EPFL, by Dubois et al. (2000). First designed in the platform Labview, it has since been further developed in the VB.net language (Hernández et al., 2007; Jordan et al., 2008). The hydrological basins and hydraulic structures are modeled by using a semi-distributed schema for free surface

flows (Hernández et al., 2009; Jordan, 2007). The model consists of four groups of reservoirs: the snowpack, the glacier, the soil and the surface runoff, which interact to produce the stream runoff. The discharge is propagated downstream by a kinematic wave relationship (Hernández et al., 2009). The meteorological inputs are downscaled by the model via spatial interpolation of the network of meteorological stations. To account for more-complex orographic phenomena that occur in mountain areas, the model divides the catchments into altitude bands of 300 meters (Hernández et al., 2009) and determines

the climatology (Jordan, 2007) with an inverse square relationship based on the distance of the barycenter of the band to stations within a certain radius. The tuning of the temperature and of the snow melt variation is therefore made more accurate through the calibration of the radius, the precipitation and temperature gradients, the degree day of the snow pack, a.s.o. RS is currently used and further developed and expanded to produce operating forecasts (hydropower production, flood protection, urban systems at Hydrique Engineers, Mont-sur-Lausanne, 1052, Switzerland). The Center for alpine

environment research (Crealp) in Switzerland also uses an operating version of RS, called MINERVE, for flood forecasting and management in the Upper Rhone River.

### 3.1 Real-time simulation and initial conditions

For the calibration of the real-time simulation, the period 2003−2015 was used for the Arve River, and 2000−2014 was used for the Rhone River. The models were calibrated over the full periods, in order to quasi-eliminate the error due to

initialization and to ensure a good evolution of the glacial surfaces (corrected within the model). The discharge gauging stations from the Federal Office of Environment were used to calibrate the model and compute the indicators (Table 1). For





the Arve River, the difference between the measured and real-time simulated volumes over the considered period was, on average, 5.2 %, and it was 0.4 % for the Rhone River.

Every scenario is initialized by the real-time simulation on the first day of the month (March to October) for lead times of 30, 90 and 120 days.

## 3.2 Past climatology and meteorological input

The seasonal forecast is forced with an ensemble of historical climatology. The climatic scenarios correspond to the measured data from the same years as the real-time simulation (2003−2015 for the Arve River and 2000−2014 for the Rhone River). The past climatology used as a meteorological input was obtained from the Federal Office of Meteorology and Climatology (MeteoSwiss) and the MeteoFrance network (see Fig. 1 for their locations). The MeteoSwiss stations were available with an hourly time resolution. The MeteoFrance stations were acquired with a daily maximum and minimum temperature and with daily cumulative values for the precipitation. The daily values are disaggregated into hourly values by using the nearest reference station where hourly data are available. Hourly precipitation from the reference stations are scaled to match the daily cumulative precipitation at the MeteoFrance station. Similarly, the hourly temperature signal from the reference station is adapted so that the minimum and maximum temperature match the ones from the MeteoFrance station. In the Arve catchment, only MeteoFrance stations were available, whereas for the Rhone, MeteoSwiss precipitation gauges were used in addition to the meteorological stations. The historical forcing remained the same for each initialization, which means that the meteorological forcing was not performed in a sliding window fashion. For example, an initialization on the first of March 2003 will produce 12 scenarios from the climatology of March 2003−2015. This is the same for an initialization on the first of March 2008, where the scenarios will be generated with the climatology from 2003−2015. The main difference between the first of March 2003 and 2008 is the soil saturation content and the snow height given by the real-time simulation.

## 3.3 Performance

The performance of the forecast is often assessed by a leave-one-out cross-validation method. One year (or several) of meteorological data is chosen to calibrate the model. The other years are kept to be used as forcing to produce the forecast (Robertson et al., 2013), making the forecast statistically independent. In this study, due to the presence of several glaciers, the whole period was used to calibrate the model (Robertson et al., 2013). Thus the leave-one-out cross validation could not be used. The performance is given by indicators that are based on the quantiles calculated from the volume of the forecasted scenarios (Fig. 4), which indicate the ability of the forecast to reproduce the volume observed and its range but not its evolution with time. The proposed two indicators are the "thinness" and "accuracy". Each indicator assigns a score to the forecast initialized at the beginning of the month (March to October) and each lead time. Consequently the indicators defined in Eq. (3), (4), (5), (6), (7) and (8) are calculated for each month of initialization, for lead times of 30-, 90- and 120-day.





These different lead times are represented by the $T^{-1}$ in the units of the cumulated volumes and can vary in days depending on the select lead time.

The percentiles used to calculate the indicators were calculated as follows:

$$h = (L + 1) \cdot p, \tag{1}$$

$$Q_p = x_{\lfloor h \rfloor} + (h - \lfloor h \rfloor) \cdot (x_{\lfloor h \rfloor + 1} - x_{\lfloor h \rfloor}), \tag{2}$$

where $h$ is the rank, $L$ is the length of the vector, $p$ is the percentile, $Q_p$ is the percentile of interest ($m^3\,T^{-1}$), $\lfloor h \rfloor$ is the floor function of $h$, and $\boldsymbol{x}$ is the vector of cumulated discharge ($m^3\,T^{-1}$).

The accuracy ($A_{ccu}$) per month is defined as the correlation coefficient R between the median volume of the cumulated scenarios and the volume observed across the years for a certain month of initialization (the volumes were computed over a certain lead time). The maximum score of 1 indicates perfect accuracy. For a score over 0.6, the accuracy is very good, and for scores between 0.5 and 0.6, the accuracy can be considered better than the average. A score lower than 0.5 reveals that the forecast does not have a linear relationship with the measurements.

$$A_{ccu} = \frac{\sum_{i=yr}^{N}\{(v_i - \bar{v}) \cdot (o_i - \bar{o})\}}{\sqrt{\sum_{i=yr}^{N}(v_i - \bar{v})^2} \cdot \sqrt{\sum_{i=yr}^{N}(o_i - \bar{o})^2}}, \tag{3}$$

where $v_i$ ($m^3\,T^{-1}$) is the median of the volumes produced by the scenarios for the chosen *month* and *year i* of the initialization and a defined lead time, with $N$ the number of years. $\bar{v}$ ($m^3\,T^{-1}$) is the average of all $v_i$ over the $N$ years. $o_i$ ($m^3\,T^{-1}$) is the observed volume for the chosen *month* of *year i* and the defined lead time. $\bar{o}$ ($m^3\,T^{-1}$) is the average of $o_i$ over the $N$ years.

The thinness ($T_{hin}$) evaluates the spread of the 80 % confidence interval (CI) of the forecast in comparison with the 80 % confidence interval of past measurements. It corresponds to the ratio between the average size of the 80% CI of the scenarios and the average size of the 80% CI of the measurements and is defined as:

$$T_{hin} = 1 - \frac{\frac{1}{N}\sum_{i=yr}^{N}\{V^i_{p=0.9} - V^i_{p=0.1}\}}{\frac{1}{N}\sum_{i=yr}^{N}\{O^i_{p=0.9} - O^i_{p=0.1}\}}, \tag{4}$$

where $V^i_{p=0.9}$ ($m^3\,T^{-1}$) is the 0.9 percentile of the volume simulated over a chosen lead time out of all the scenarios for a *month* and *year i* of initialization, and $V^i_{p=0.1}$ ($m^3\,T^{-1}$) is the percentile 0.1. All the percentiles are calculated with Eq. (1) and (2). $O^i_{p=0.9}$ ($m^3\,T^{-1}$) is the 0.9 percentile of the volume observed over a chosen lead time out of all the measurements for the chosen *month* of *year i*.





For example, if the thinness is 5 %, it means that the 80 % CI of the forecast is on average 5 % thinner than the 80 % CI of past measurements. The forecast is categorized as thinner for a score higher than 5 %, as similar between -5 % and 5 % and as larger for scores lower than -5 %.

**3.4 Mean absolute error**

To gain a better understanding of the performance of the average of past year measurements, the NMAE was introduced. It describes the overall error committed by the average of past year measurements or the seasonal forecast compared with the actual measurements of the period of interest for a certain initialization. Its mean absolute error (MAE or $E_{MA}$) was compared with the MAE of the seasonal forecast and normalized (NMAE or $E_{NMA}$) with the observed values $\bar{o}$.

Evaluating the NMAE is important because it quantifies the error on the volume at the end of the simulation.

The MAE for the month of March, for example, is based on the absolute difference of the median volume produced after a certain lead time for a specific year and the measured volume during that same year.

$$E_{MA} = \frac{1}{N} \cdot \Sigma_{i=yr}^{N} \{|v_i - o_i|\}, \tag{5}$$

$$E_{NMA} = \frac{E_{NMA}}{\bar{o}}, \tag{6}$$

**3.5 Error inherent to the calibration**

To quantify the error within the seasonal forecast due to the calibration, the normalized mean error (NME or $E_{NM}$) calculated and rescaled based on the average volume observed $\bar{o}$ to obtain a percentage for each month and horizon.

The mean error (ME or $E_M$) of the real-time simulation was determined on the days of initialization and after the different lead times.

$$E_M = \frac{1}{N} \cdot \Sigma_{i=yr}^{N} \{s_i - o_i\}, \tag{7}$$

$$E_{NM} = \frac{E_M}{\bar{o}}, \tag{8}$$

where $s_i$ (m³ T⁻¹) is the volume of the real-time simulation produced between the initialization and up to a chosen lead time. ESP forecasts tend to be biased due to inadequate model parameterization or other errors (Mendoza et al., 2017). Indeed, the

seasonal forecast of the Rhone River was biased. Therefore, the results were corrected according to the volume bias between the real-time simulation for the different lead times and the measurements. The average of the volume produced by the real-time simulation was divided by the average of the volume measured over the same lead time for each month of initialization, resulting in a correction factor that was applied to the forecasted volume as a division factor.



# 4 Results

In the following sections, the results will be presented, first for each of the rivers separately and then by a joint view on the seasonal forecast. The performance of the forecast for each river is assessed by indicators of thinness and accuracy, which can be found in Fig. 5, where they are color coded to help interpretation. The scales go from green to purple for the accuracy and from green to pink for the thinness. The threshold of significance is white, which for the accuracy corresponds a range of 0.45−0.55 and for the thinness of -0.5−0.5 %. Above the thresholds, the forecast is considered to be better than the average of past measurements. The exact scores of the performance indicators can be found in the supplementary material in Tables S1 and S2.

The comparison between the NMAE and the performance indicators validates the use of the seasonal forecast for a certain initialization and lead time. The NMAE results can be found in Fig. 6 and Fig. 7. The NME results, which were used to correct the bias on the Rhone River are presented and discussed in Appendix A: Real-time simulation error (Fig. A1 and Fig. A2).

## 4.1 Arve River

### 4.1.1 Post-processing the forecast

Figure 5 shows the different results, including some that are labeled biased, referring to indicators that needed to be corrected. The correction was needed due to two floods that occurred on the 3rd and 9th of July, 2007. The volume produced by the floods of 2007 could not be reproduced by the meteorology of other years. The accuracy was thus weighed down by this outlier. To improve the accuracy, the forecast of 2007 was excluded from the calculation for 90 and 120 days lead time for all months. The thinness was not recalculated without the forecast of 2007, due to its nature of measuring the spread of the forecast including the extreme events.

### 4.1.2 Performance of the seasonal forecast in terms of initialization and lead time

Looking at the corrected indicators, it can be seen that different lead times can be more favorable for different months of initialization. For the initialization in March, July and August, the accuracy was best for a 120-day lead time, but the thinness was best for 30-, 90- and 120-day lead times, respectively.
For initialization in April and June, the most favorable lead time was 30 days, while for initialization in May and July, the forecast was best for 120 days. Initialization in October should be reliable for 90 days lead time, but the other lead times were just under the threshold of significance (0.47 and 0.45 for the 120- and 30-day lead times respectively) in terms of accuracy. On the other hand, forecasts initialized in March, August and September did not provide a reliable forecast. Additionally, it was seen that a reliable forecast will come out of initialization in April, May and June for all lead times and in July for lead times of 30 and 120 days. The thinness was also good in July for 90-day lead times, but the accuracy was





0.45 for the corrected forecast. In this case, the exclusion of the 2007 forecast negatively impacted the accuracy, which was 0.5 without the post-processing. Thus, the forecast initialized in July was still considered usable for all lead times. The initializations in September did not have sufficient scores to provide a good forecast, with no linear relationship to the measurements and relatively large thinness.

### 4.1.3 Performance of the seasonal forecast in comparison with the multiannual predictand

The indicator analysis gave good information on the behavior of the median of the seasonal forecast and its 80% confidence interval compared to the measurements, but the indicators did not hold information on the absolute error committed by the forecast in comparison with the average of past year measurements (multiannual predictand). The comparison between the NMAE of the seasonal forecast and the multiannual predictand revealed the forecast for which the smallest absolute error, on average, for the measurements was obtained (Fig. 6).

For the Arve River, the error of the seasonal forecast was smaller than the multiannual predictand for initializations from March to July across all lead times with or without the forecast of 2007. The 30-day lead time showed the highest number of cases where the seasonal NMAE was larger than or equivalent to the multiannual predictand. Similarly to the performance computation, the seasonal NMAE, without the 2007 forecast, was reduced for the month affected by the floods. The 120-day forecast was the most influenced by the deletion of the 2007 forecast.

### 4.2 Rhone River

### 4.2.1 Post-processing the forecast

Looking at the real-time simulation of the Rhone River, large biases were obtained for different years and months up to ± 20 %. To correct the bias, the differences in volume between the real-time simulation and the measurements were averaged, and the seasonal forecast was corrected accordingly. Correcting the forecast of the Rhone River improved the thinness of the forecast for all months and all lead times, except for the month of March with a 30-day lead time, where the correction increased the spread of the 80 % CI. On average, the real-time simulation tended to be lower than the measurements in March, but extreme cases such as 2009, when the real-time simulation was much lower, strongly influence the average bias. The correction of the forecast was thus too important and led to a larger spread, decreasing the thinness.

### 4.2.2 Performance of the seasonal forecast in terms of initialization and lead time

The initializations in April, May and July showed the best reliability for the 120-day lead time, while the initialization in June was the best for a 30-day lead time. The initialization in April for the 30-day lead time obtained good accuracy, but the thinness was large. Nevertheless, the forecasts seem reliable due to the good correlation of the median forecast with the measurements. This increased the probability that the measured value would be inside the 80 % CI. From August to March, the initializations did not have a linear relationship with the measurements and were not reliable. All the forecasts initialized





from April to July were reliable for all lead times, and the 120-day lead time showed the best scores, with high accuracy and good thinness. Thus, the score increased with increasing lead time, which seemed counterintuitive, since in most cases, forecasts tend to be better at shorter lead times.

### 4.2.3 Performance of the seasonal forecast in comparison with the multiannual predictand

In terms of NMAE, correcting the bias allowed one to increase the number of cases where the seasonal NMAE was smaller than or in a similar range as the multiannual predictand (Fig. 7) (favorable cases). The initialization for which the corrected seasonal forecast gave smaller errors is from April to August for all lead times. The number of favorable cases also increased with lead time. Post-processing increased the number of favorable initializations of the seasonal forecast, highlighting the need to post-process the forecast. The supplementary information given by the NMAE confirmed the use of the seasonal forecast for initialization in April for a 30-day lead time, although its thinness was poor.

## 5 Discussion

For both rivers, the accuracy and thinness showed different optimal lead times, which made it difficult to assess which lead time was the most favorable. Joint information given by both indicators should be extracted in order to define the lead time that provides the best forecast. This means that both indicators should be above 0.5 for the accuracy and more than -5 % for the thinness.

Based solely on the indicators, the initializations in August on the Arve River had a linear relationship to the measurements and were almost usable. While October showed good accuracy and good thinness for a 90-day lead time, the other lead times had somewhat poor accuracy, making it difficult to assess whether the initialization in October would have provided a reliable forecast. Additional information was taken from the NMAE analysis to determine which lead times were usable. Across all lead times, the months of initialization that showed a smaller seasonal NMAE stay nearly the same, except for the month of October, which showed a similar performance to the multiannual predictand for the 30-day lead time but did not have satisfying indicators. The month of October could therefore be excluded from the reliable initialization of the seasonal forecast. March had a relatively low reliability in terms of indicators but a smaller error in terms of NMAE, which was not seen for other initializations. This shows that initializations in March will, on average, show smaller errors than the multiannual predictand, but the use of the forecast remains risky in terms of indicators.

On the Arve River, the exclusion of 2007 increased the accuracy of the 90- and 120-day lead times of all the initializations, except for the initialization in July and August, where the accuracy decreased. This decrease in accuracy is due to the missing forecast of 2007 whose volume compensated for the droughts of August 2009, 2010, 2011 and 2015.





The counterintuitive increase in indicator scores with lead time found on the Rhone River can be explained by the following, since the skill of the model in the size of the snow pack is high, the skill in the timing of the melt is low. The low skill in melt timing is due to the fact that the climatology from previous years is used in the prediction. For this reason, having a good estimation on the flow volume over a month is more difficult than having a good estimation over four months, during

which the melt will certainly occur. Moreover, the purpose of the calibration was to obtain a volume similar to measurements over the full year. Thus, the calibration also helped compensate for the differences over time. Additionally, the diurnal pattern due to hydropower production was difficult to assess, so the longer the lead time, the more likely it was that the volume would be produced during the period.

The soil moisture content strongly contributes to the run-off over the autumn and winter months, when the melt water decreases and the rainfall and base flow become more important. On top of the base flow and the contribution of the rainfall, the proportion of the flow due to hydropower production will increase over these months. No skill in autumn and winter for either catchment was identified. Calibration during these months was more difficult due to the presence of the hydropower installations, especially on the Rhone River, where they are numerous. Looking for a skill for months where the soil moisture

content plays a major role might be more interesting on rivers with a pluvial regime (as defined by Musy, n.d.). For high-altitude catchments, the proportion of the soil water content over the winter months is the main or even the only contributor to the flow. Thus, a skill in winter might be identifiable, but it might not be very interesting because of the lack of interest of the actors for the low flows. For autumn and winter months, the average of past years measurements is preferable.

The advantage of ESP is that it takes into account a "nowcast", potentially leading to a skill in seasonal forecasting. The initial condition in spring greatly depends on the height of the snowpack and glacial storage, which are the main drivers of the flow over the summer months. A skill in spring was expected for pluvio-nival catchments due to their flow regime. The results revealed that the snowpack plays a key role in the predictability for all lead times of both catchments over the summer months for initializations from April to July.

The identified skill depended on the snow and glacial storage. Therefore, errors in the meteorological forcing will cause larger errors in the forecast. The simulations initialized from April to July were typically influenced by the melt water, which depended on the size of the winter storage and, thus, the forcing. To address this error, the post-processing of the forecast was useful on the Rhone River and allowed its skill to be extended. On the Arve River, the error of the seasonal forecast was

decreased compared with the multiannual predictand, as shown through the NMAE analysis. Just as expected in ESP, the initial conditions were not perfect, but the post-processing of the forecast can improve some indicators, as shown by Crochemore et al. (2017). Even though the forcing used was a seasonal meteorological forecast, the results were similar in the way that they improve the indicators, but they did not improve all of them at the same time. Thus, the performance was enhanced, but too many modifications could hinder the overall reliability of the forecast. Additionally, post-processing of the





forecast is possible, but it is more time consuming because it needs to be applied to each forecast run in the operational platform.

The representativeness of the forcing is an important factor in the quality of the seasonal forecast. The problem with the floods occurring in 2007 revealed the dilemma regarding the choice of forcing and its impacts on the scores of the forecast. Adapting the forcing set to calculate the predictand is feasible. It can be shown that the median needed to be considered without 2007 to obtain better accuracy, while the spread should include the extremes. In further developments, two sets of meteorology could be considered to build the forecast, one including "normal" years to compute a strong median and a

second with more extreme weather events, to improve the description of the spread.

Seasonal forecasting gives the most probable scenario based on the "nowcast" for average meteorology over the coming months. While it is interesting to see the propagation of these initial conditions with time, the forecasting of extreme events such as floods will remain low in probability. For example, a small snow pack will produce lower flow conditions during the snowmelt period, with average meteorology. Similarly, a larger snow pack will produce higher flow over the summer. The

occurrence of floods depends on the base flow conditions and the meteorology. While the probability of a larger base flow will increase for a larger snowpack, the probability of the extreme meteorology will remain low. To predict floods or, inversely, drought events, specific forcing should be used.

The applied method does not ensure that the calibration is independent of the forecast; however, it does give interesting

results. In the operational setting, the calibration is controlled and adapted if necessary. Additional future reanalysis will include more meteorological measurements, which will be added to the ensemble of climatology for the seasonal forecast, making the forecast independent of the calibration period.

The accuracy and thinness scores were high for these models and are easy to communicate. For some cases, it is difficult to

combine the information of both indicators to conclude on the validity of the forecast. The NMAE gave deeper insights into the overall validity of the forecast, but it might be more difficult to communicate, due to its nature of ensuring an average margin of the median member of the forecast over a decade. The range of validity was the same for both rivers, from April to July, with the bias correction applied. The initialization in March showed potential in terms of NMAE, with accuracies slightly under the threshold, though acceptable, which lefts potential in the extension of the range of validity.

**6 Conclusion**

The skill of the ESP method was tested on the Rhone and Arve rivers in comparison with the performance of the statistical predictand (average of historical measurements) for 30-, 90- and 120-day lead times. For both rivers, the seasonal forecast

had potential for initializations starting from April to July. For the rest of the months, the statistical predictand were preferred in the current state of knowledge. The Arve showed the classical decrease in skill with increasing lead time, but the accuracy remained within a good range, even for the 120-day forecast. On the Rhone River, the relationship to the lead time was interestingly reversed due to the higher probability of a larger lead time to obtain the correct timing of the melt water.

Overall, a skill was found in spring and summer during the snow melt period for pluvio-nival catchments such as the Arve and Rhone rivers. Consequently, the seasonal forecast can be used from April to July for lead times up to four months for both rivers, as was validated by the good accuracy and the satisfying spread of the 80 % CI. The addition of the normalized mean average error (NMAE) as an indicator of the seasonal forecast provided relevant information on the range of validity of the forecast. The soil moisture content, as opposed to the snow pack height, does not dominate the discharge pattern for long

lead times. The rainfall and daily hydropower production become larger than the baseflow during the autumn and winter months, meaning that for forecasts initialized in August, September and October, the average of the past measurements should be preferred. The use of post-processing to correct errors in the forecasts was needed, and it helped decrease the seasonal NMAE on the Arve River and increase the number of favorable initializations in terms of NMAE on the Rhone River. Further development should investigate the possibility of dividing the meteorological forcing into two sets, one to

have a solid median forecast and a second with extreme cases to have a solid confidence interval. Optimization of the autumn and winter months via model improvements needs to be researched.

**7 Code availability**

The Routing System model used in this contribution is developed by Hydrique Engineers and is proprietary. There is a free Routing System version, called RS MINERVE that can be downloaded at the following address:

http://rsminerve.hydro10.org/download.

**8 Data availability**

Available research data are presented in the Supplement. The meteorological data belong to MeteoFrance and MeteoSwiss and cannot be shared by the institute. The historical discharge data of the gauging station can be found at a daily resolution at the following address: https://www.hydrodaten.admin.ch/en/2009.html.

**9 Appendices**

**Appendix A: Real-time simulation error**

The real-time simulation error was evaluated to obtain information on the errors inherent to the calibration. For the Arve River, the NME was not constant with initialization. It varied from positive to negative and in size, meaning that the model

was not monthly biased. The error oscillated around 5 % for the whole initialization. Similarly, the NME varied with increasing lead time.

The largest initialization error for the Arve River was obtained for the months of June and September. Overall, the NME of the Arve River's real-time simulation was mostly within the range of -10 % and 10 %. The relatively small NME showed a good calibration of the model.

The error on the day of initialization of the 1st of March on the Rhone River was the largest, at approximately 35 %. Over the first month, the error jumped down to -15 %. This phenomenon illustrates the behavior of the calibration, where over a certain period of time, it can overcompensate for the error in initialization. For the Rhone River, the error for 90 and 120 day lead times was under 15 %. The error on the day of initialization was greater than 10 % from March to June and in October. For the Rhone River, the NME varied with each month and lead time, but it was usually positive. This overly positive NME highlighted the tendency of the calibration to exceed the measured volumes. The indicators were unbiased accordingly. Correcting the result for the Rhone brought the NME close to zero for all initializations and lead times. This illustrates the complexity of the Rhone, which was more difficult to model and calibrate than the Arve River.

## 10 Supplement link

### S1 Performance indicators

The detailed values of the performance indicators are given in Tables S1 and S2.

## 11 Team list

Oriane Etter, Frédéric Jordan, Anton J. Schleiss

## 12 Author contributions

Frederic Jordan and Oriane Etter designed the experiments, and Oriane Etter carried out the simulations and prepared the manuscript with the contribution of all co-authors. Preliminary tests of this method were performed by Dr Guillaume Artigue and Alexis Laulagnet at EPFL and Hydrique Ingénieurs.

## 13 Competing interests

The authors declare that they have no conflict of interest.





## 14 Acknowledgments

The developments are part of the OPT-HE research project CTI 16124.1 PFEN-IW. Industrial partners of this project are MeteoSwiss, Alpiq, the Service Industriel de Genève (SIG), the Groupe-e, Romande Energie SA, Société Electrique des Forces de l'Aubonne, and IACETH.

The authors thank Damien Puygrenier and Marc Bernard from EDF for their advice in defining the performance indicators.

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



**Figure 1: Study domain and meteorological stations**





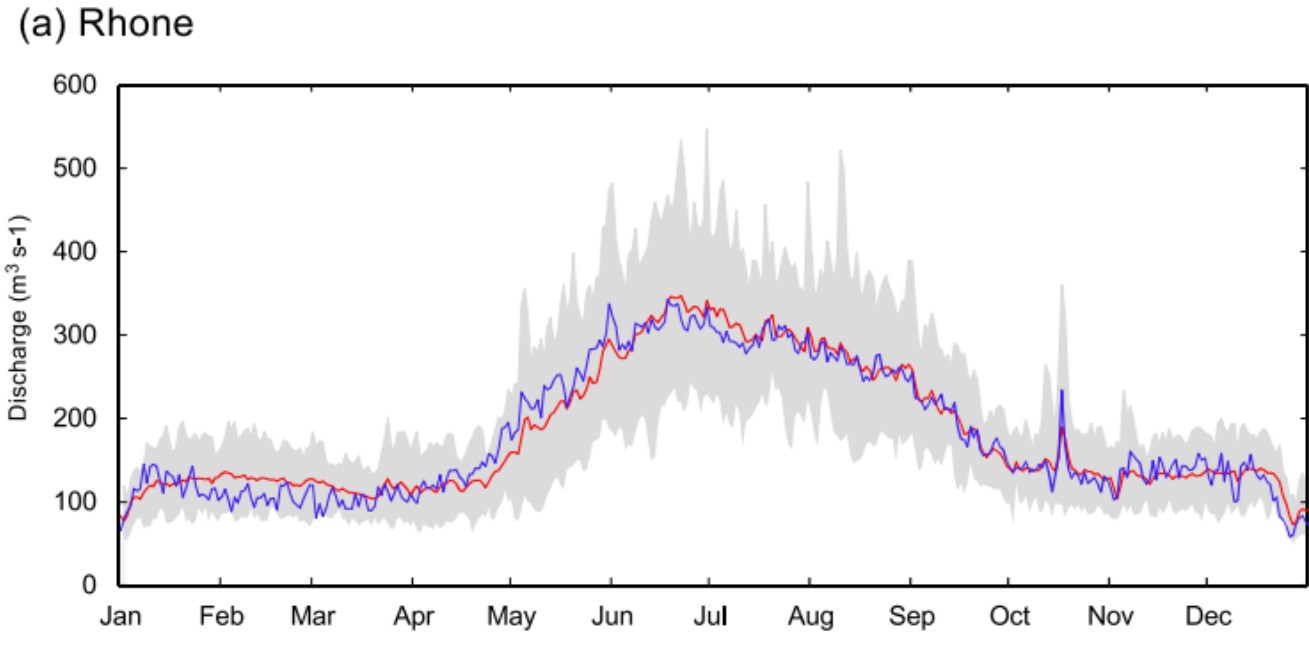

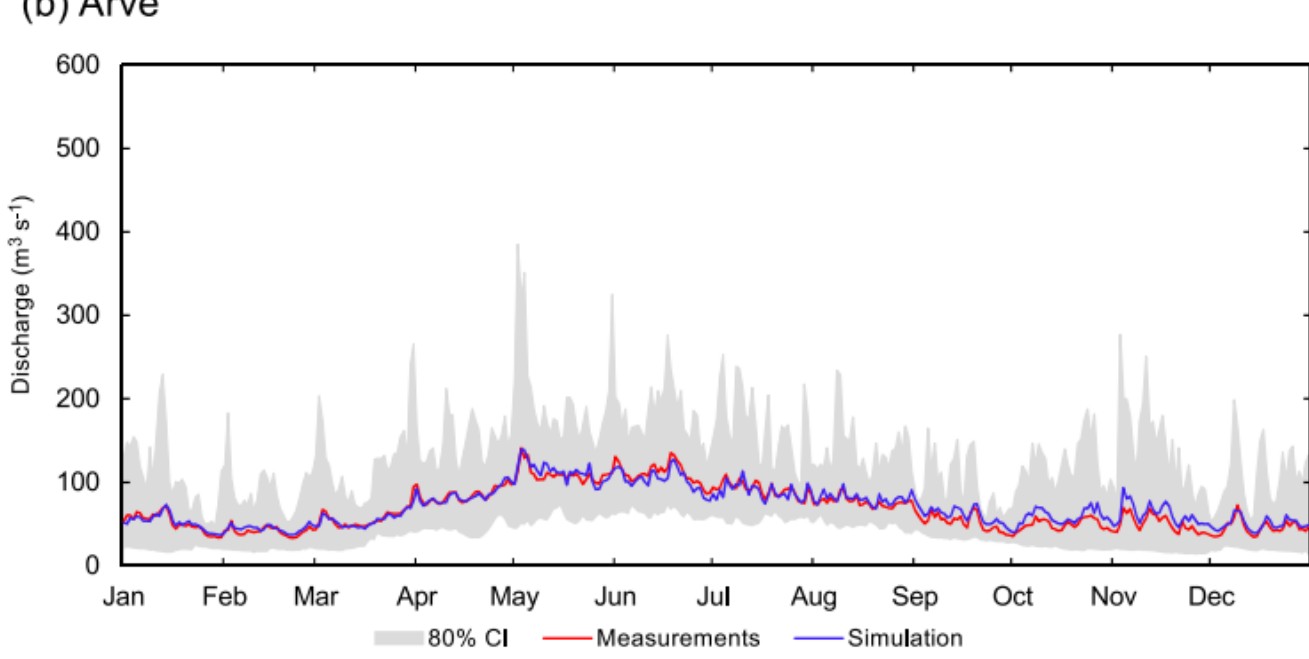

**Figure 2: Daily average discharge over the validation years for the past measurements and the simulation. The 80 % confidence interval is computed based on the percentile of the past year's measurements over the calibration period.**



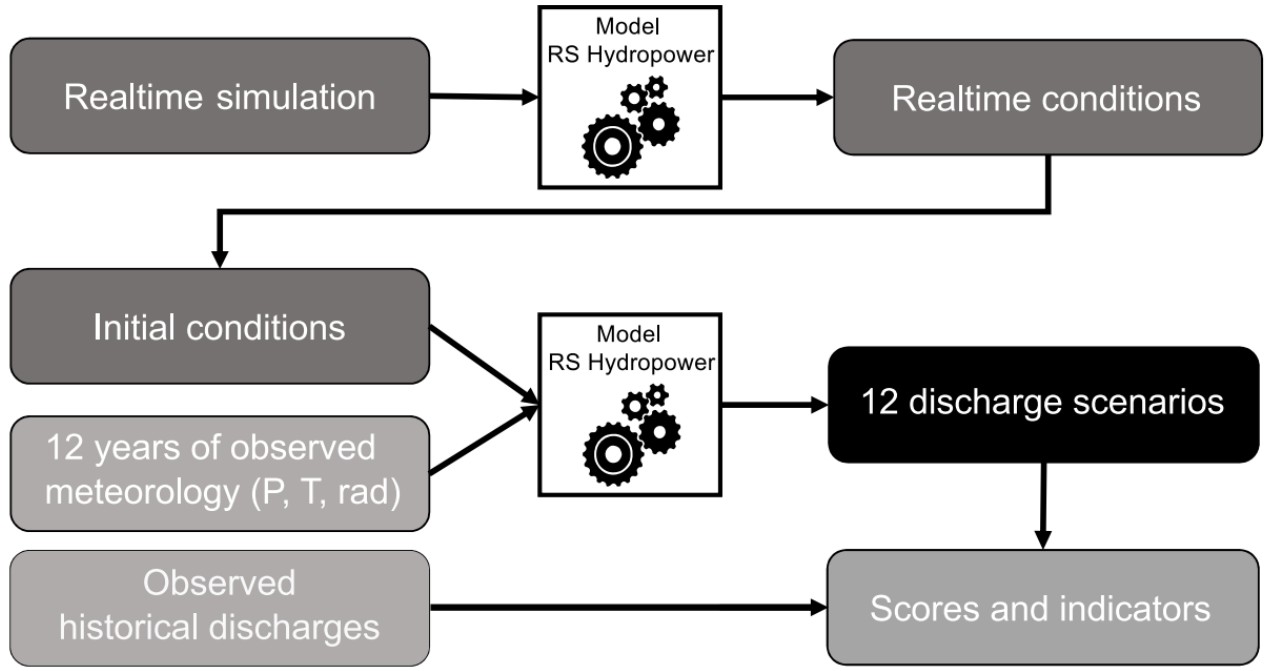

**Figure 3: Seasonal forecasting methodology: The Routing System model runs the real-time simulation on which the seasonal forecast is initialized. Past climatology is used to produce an ensemble of discharge scenarios, from which performance indicators and scores are computed.**

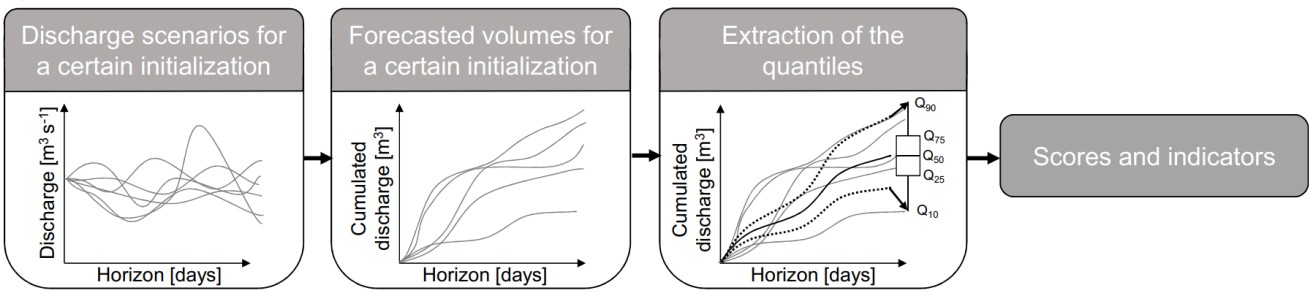

**Figure 4: Illustration of the methodology to pass from the scenarios to the scores and indicators: The Routing System model computes one scenario per year of past climatology available (*n* years), resulting in *n* discharge scenarios being produced. The discharges of these scenarios are cumulated, and quantiles are extracted from them. These quantiles are used to calculate the scores and indicators.**





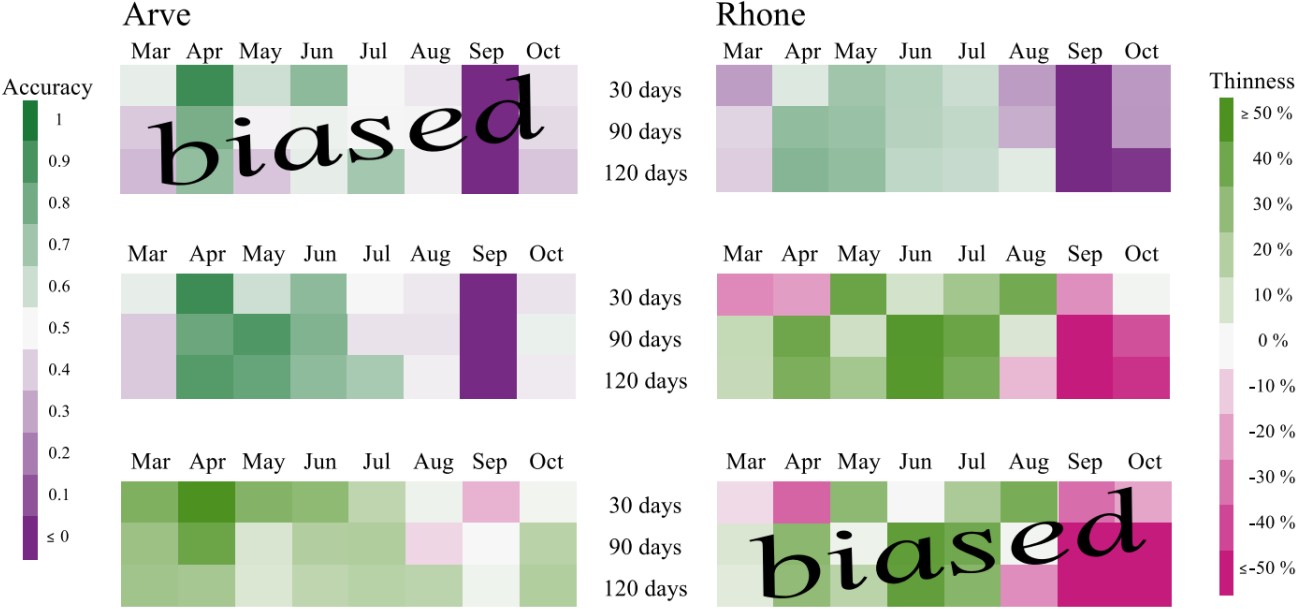

**Figure 5: Performance indicators for the Arve and Rhone rivers.**

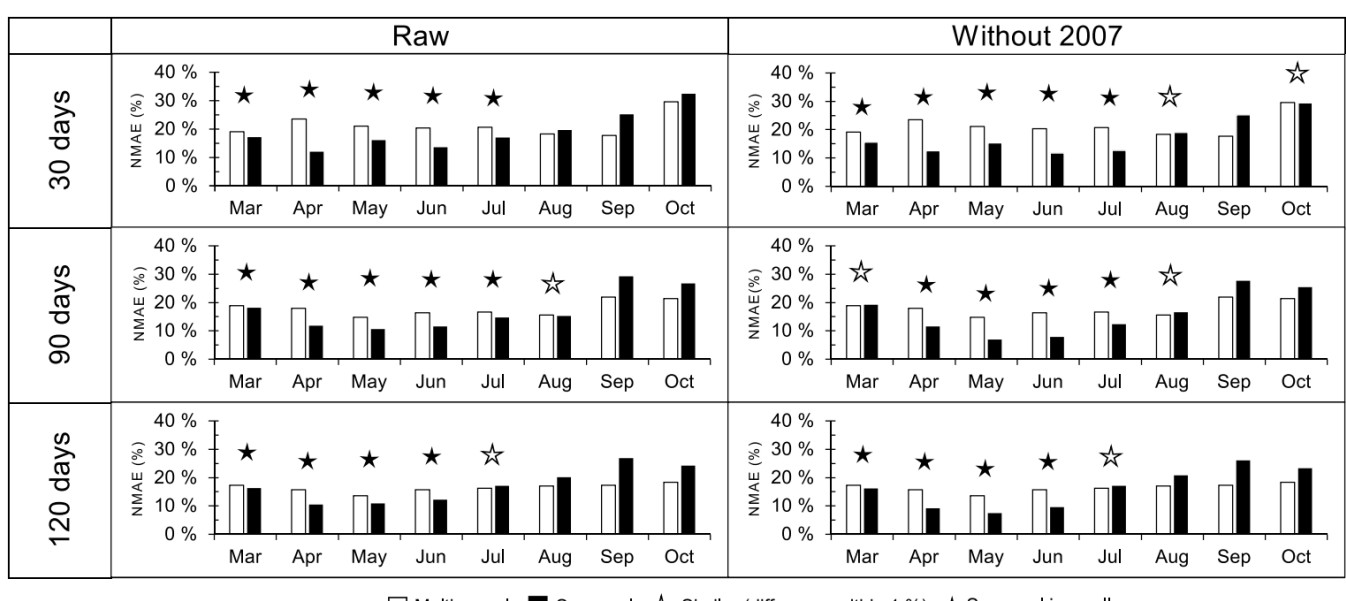

5 **Figure 6: NMAE of the seasonal forecast and multiannual predictands for the Arve River.**





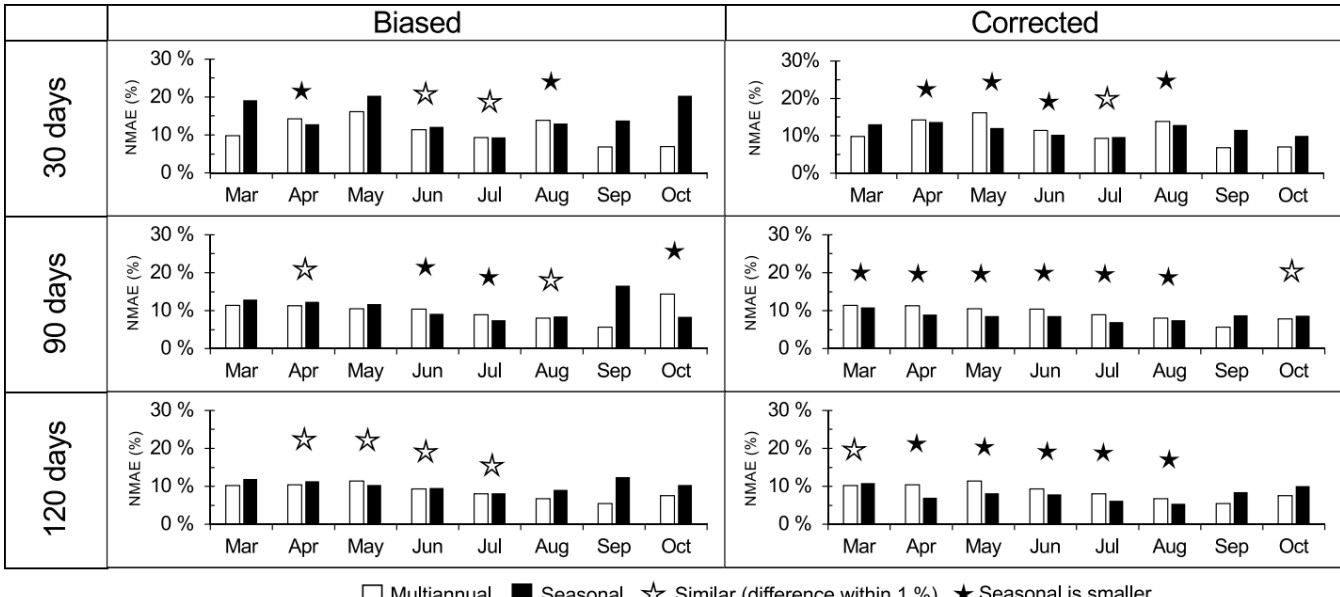

**Figure 7: NMAE of the seasonal forecast (in Black) and multiannual predictands (in White) for the Rhone River.**



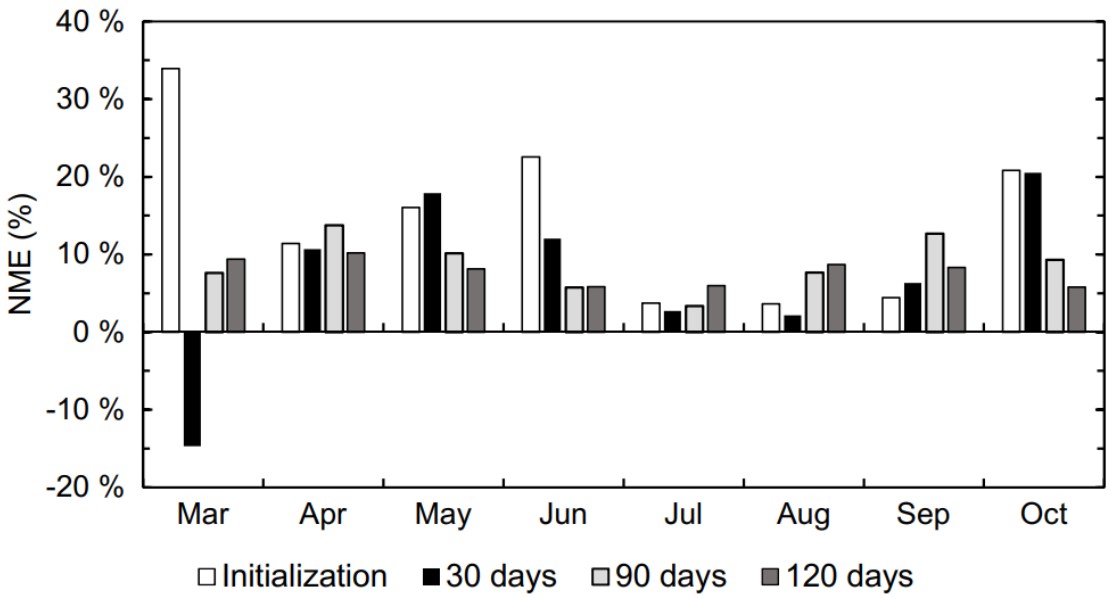

**Figure A1: Normalized mean error (NME) of the real-time simulation of the Rhone River**

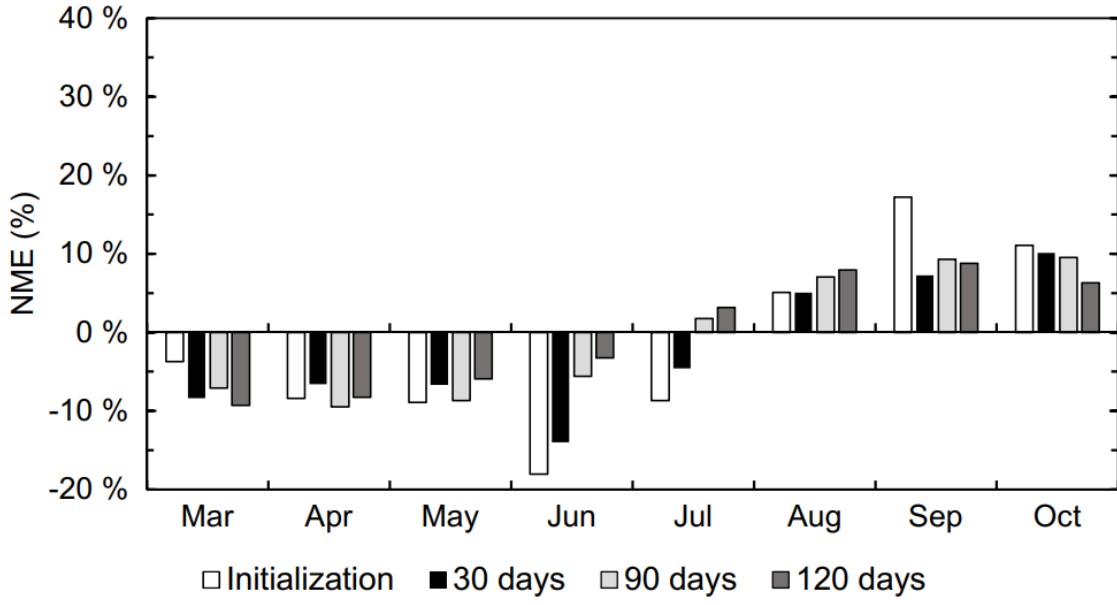

**Figure A2: Normalized mean error (NME) of the real-time simulation of the Arve River.**





| Basin | Gauging station | Area (glacial surface) ($km^2$) | $Q_{avg}$ ($m^3\ s^{-1}$) | Sources |
|-------|-----------------|-------------------------------:|-------------------------:|---------|
| Arve  | Genève-Bout du monde | 1930 (75) | 80 | (Francou and Vincent, 2007; Grandjean, 1990) |
| Rhone | Porte du Scex | 5500 (700) | 180 | (Grandjean, 1990; Hernández et al., 2007, 2009) |

**Table 1: Basin characteristics and gauging stations.**