# Peer review of "Potential of seasonal hydrological forecasting of monthly run-off volumes for the Rhone and Arve Rivers from April to July"

_Hydrology and Earth System Sciences, 2018_

## Referee Comment (RC1) · Anonymous Referee #1 · 9 Apr 2018

This manuscript on seasonal forecasting does not present any significant methodological advance. It might after major revisions qualify a manuscript type called "Cutting-edge case studies", which according to HESS standards can be published if they " report on case studies that require (a) broadening the knowledge base in hydrology as well as (b) sharing the underlying data and models." https://www.hydrology-and-earth-system-sciences.net/about/manuscript_types.html

These two conditions will probably not be met for this paper and accordingly, I do not think that this paper is within the scope of HESS.

Given the above, I do not provide a full review at this stage. I have nevertheless the

following detailed comments:

- The paper does not sufficient details about the used model. It refers to old papers but from the reading of the paper I get the impression that the model is used in real engineering applications, meaning that it has certainly undergone significant development since its original publication

- The model apparently contains some glacier surface evolution module, which is not mentioned in the model description section ("the models were calibrated over the full periods, in order to quasi eliminate the error due to initialization and to ensure a good evolution of the glacial surfaces (corrected within the model)."

- Model calibration is mentioned many times in the results section but not discussed in the methods section. As far as I see, model calibration is just mentioned as "The tuning of the temperature and of the snow melt variation is therefore made more accurate through the calibration of the radius, the precipitation and temperature gradients, the degree day of the snow pack, a.s.o". This is not sufficient by any means.

-The method section mixes methodological descriptions and details about the case study data and results

---

## Referee Comment (RC2) · Anonymous Referee #2 · 26 Apr 2018

Review

This paper presents some interesting forecast examples, but I doubt it is suitable for HESS for the following reasons:

- The writing is very unclear. What is written often does not make sense. I recommend that a native speaker checks the entire manuscript. - It is unclear what the real novel contribution is of this paper. I would expect statements in the abstract or conclusion that would make the scientific contribution of this paper clear. Instead, I only read details about how models have performed, without any context of what we actually learn from these results, and how that improves our knowledge compared to before

this manuscript was written. I also do not see how there are any clear significant methodological advances. - The methods section mixes methods and results, and data. - The study domain section provides results - Similar to the previous reviewer: details about the used model are insufficient and from the reading of the paper I have no idea what model is actually used in the end. I cannot reproduce any of the methods. - Please provide figure captions that actually explain the figure. - I agree with the previous reviewer that given the quality of this manuscript I cannot provide a full review of the content presented in this paper.

Sorry I cannot be more positive about this paper, but I fail to see how it can become an appropriate contribution to HESS.